# Estimation of Stiffness of Non-Cohesive Soil in Natural State and Improved by Fiber and/or Cement Addition under Different Load Conditions

**DOI:** 10.3390/ma16010417

**Published:** 2023-01-01

**Authors:** Katarzyna Zabielska-Adamska, Patryk Dobrzycki, Mariola Wasil

**Affiliations:** Faculty of Civil Engineering and Environmental Sciences, Bialystok University of Technology, ul. Wiejska 45E, 15-351 Bialystok, Poland

**Keywords:** resilient modulus, California bearing ratio (*CBR*), static loading, cyclic loading, compacted soil, cement stabilization, fiber reinforcement

## Abstract

The aim of this study was to compare the stiffness of gravelly sand under various load conditions—static conditions using the CBR test and cyclic conditions using the resilient modulus test. The tests were conducted on natural soil and soil improved by the addition of polypropylene fibers and/or 1.5% cement. The impacts of the compaction and curing time of the stabilized samples were also determined. The soil was sheared during the $M_{r}$ tests, even after fiber reinforcement, so the resilient modulus value for the unbound sand could not be obtained. The cement addition improved $M_{r}$, and the curing time also had an impact on this parameter. The fiber addition increased the value of the resilient modulus. The *CBR* value of the compacted gravelly sand was relatively high. It increased after adding 0.1% fibers in the case of the standard compacted samples. The greater fiber addition lowered the *CBR* value. For the modified compacted samples, each addition of fibers reduced the CBR value reduced the *CBR* value. The addition of cement influenced the *CBR* increase, which was also affected by the compaction method and the curing time. The addition of fibers to the stabilized sample improved the *CBR* value. The relationship $M_{r}=f\left(CBR\right)$ obtained for all data sets was statistically significant but characterized by a large error of estimate.

## 1. Introduction

The soil and granular material embedded in the base and subbase of pavements are subjected to a large number of loads at stress levels considerably below their shear strength. Under a single load on a moving wheel, pavements mainly respond in an elastic way. Nevertheless, indivertible plastic and viscous stresses may accumulate with repeated loading [1], and the thickness of the pavement layers is of great importance for maintaining flexible pavements. The impact of repetitive loadings on pavements was explained in 1955 [2], and a new term, “resilience”, was introduced. The resilient response of granular material as a modulus of resilience (later known as the resilient modulus), as the ratio of repeated deviator stress in the triaxial compression test to the recoverable (resilient) axial strain, was first defined by Seed et al. [3,4]:(1)$$M_{r}=\frac{\sigma_{d}}{\varepsilon_{r}}=\frac{\sigma_{1}-\sigma_{3}}{\varepsilon_{r}}$$

This corresponds to Young’s modulus, but under cyclic loads, the elastic modulus can be replaced by the resilient modulus to account for the non-linearity and stress dependence under cyclic loading.

The resilient modulus of unbound materials has been most often evaluated in laboratories using a triaxial apparatus, but other methods, such as the simple shear test, the hollow cylinder test, the true triaxial test, and the torsional resonant column test, can also be applied. It should be noted that the tests should be performed in devices with the possibility of cyclic loading, because only then is it possible to assess the effect of stress and strain accumulation. Cyclic triaxial test procedures have been standardized; the most influential ones are the AASHTO T307 [5] and EN 13286-7 [6] standards. It should be emphasized that the European Standard [6] concerns only unbound mixtures; bound mixtures are still estimated by means of the static elastic modulus.

To date, many researchers have examined the impacts of numerous factors on the resilient modulus of several types of soil. The effects of the confining pressure and deviator stress, load frequency and duration, amount of load cycles, density, graining, and soil saturation were characterized in detail by Brown [1] and Lekarp et al. [7]. The main factors influencing the resilient modulus of unbound non-cohesive [4,8,9,10,11,12] and cohesive soils [13,14,15,16] are applied axial stress and confining pressure, but their impacts are different. In the case of non-cohesive soils, the resilient modulus increased slightly with confining pressure and significantly with repeated axial stress. The resilient modulus depended on the number of loading cycles and their frequency [17]. Tanimoto and Nishi [13] stated that, after a large number of repetitions, the resilient strains of silty clay reached a constant value after modifying the soil structure. Tang et al. [12] assessed the repetition amount of about one hundred cycles. The changeability of the accumulated plastic strain was bigger with a greater number of load cycles, a higher amplitude of dynamic stress, and a lower confining pressure the resilient modulus of cohesive soil with low plasticity [18,19] increased with an increase in matric suction and relative compaction.

Nowadays, the mechanistic design methods for pavements and pavement layers require the resilient modulus of unbounded pavement layers in order to establish layer thickness and the whole system response to traffic loads. However, the resilient modulus testing procedure is considered to be complicated, and regional road laboratories do not have cyclic triaxial apparatuses at their disposal. Therefore, the relationship between the resilient modulus value and other parameters, often even available in databases, is sought. Statistical regression models can be divided based on [20]: (I) a single strength or stiffness parameter; (II) physical soil parameters and stress state; (III) a stress invariant or a set of stress invariants; and (IV) constitutive equations for the estimation of resilient modulus values based on soil’s physical properties incorporated into the model parameters in addition to stress invariants. One such single parameter of the first group is the *California Bearing Ratio* (*CBR)*.

The *CBR* test has been commonly applied in granular material and soil testing in road laboratories for about eighty years. The *CBR* parameter is defined as the ratio of the unit load (in percent), which is used so that a standardized piston may be pressed in a soil sample to a certain depth at a rate of 1.25 mm/min and standard load, corresponding to the unit load needed to press the piston at the same rate into the same depth of a crushed rock at standard compaction. In many countries, the method based on *CBR* remains the primary method of pavement design or even the recommended method for characterizing subgrades [1]. It needs to be mentioned that the *CBR* value does not reflect the shear stress caused by repeated traffic loading. The shear stress depends on several factors, none of which is fully controlled or modeled in the *CBR* test. However, *CBR* values are strongly related to compaction characteristics, so the *CBR* test can be applied as a method of assessing earthworks [21,22]. The *CBR* value is used to evaluate the subgrade or subbase penetration resistance, and it can be employed to assess the resistance to static failure.

The most frequently quoted in the literature relationship $M_{r}=f\left(CBR\right)$ is the formula given by Heukelom and Klomp [23]:(2)$$M_{r}=10\;CBR\;(\mathrm{MPa})$$

Formula (2) has been converted into SI units. It should be mentioned here that the relationship was not originally related to the resilient modulus obtained in laboratory tests but to the dynamic modulus, which was determined based on vibration wave propagation. Nevertheless, this relationship is commonly used in the form of the relationship $M_{r}=f\left(CBR\right)$. Although the regression coefficient provides the best fit with *CBR* values from 2 to 200, many researchers have limited the *CBR* values to less than 10 or 20. Many researchers have found that the relationship of $M_{r}=f\left(CBR\right)$ underestimates the $M_{r}$ value for lower *CBR* values and overestimates it for higher *CBR* values. A detailed discussion of the evaluation of the dependence of $M_{r}$ on *CBR* was carried out by Dione et al. [24]. Farell and Orr [25] confirm that Equation (2) overestimates the stiffness of fine-grained soil, especially at high *CBR* values. They believe that, at low *CBR* values, *CBR* corresponds to the stiffness of the material while at high values of its strength. An underestimation of the $M_{r}$ value of soaked gypsum sand [26] and an overestimation in the case of a soil-fly ash mixture have been found using other relationships $M_{r}=f\left(CBR\right)$ proposed in the literature [27].

The authors of this study concluded from their previous research that it may not be possible to provide the relationship of $M_{r}=f\left(CBR\right)$ for non-cohesive soil because the cyclic triaxial test can destroy unbound non-cohesive soil [28,29]. Thus, the aim of this study was to compare the stiffness tested under different load conditions in a typical non-cohesive soil used for road base and subbase construction—postglacial gravelly sand. Static and cyclic loads, represented by *CBR* and resilient modulus tests, were considered. The soil was tested as unbound and hydraulically bound by the addition of 1.5% cement, choosing the minimum amount that could improve the resilient characteristics of the tested soil. As the authors’ previous experiences have shown that non-cohesive soil can be sheared during the cyclic triaxial test, it was also decided to test soil samples reinforced with 18 mm-long polypropylene fibers to improve the tested material [30]. Thus, the behaviors of four different materials were considered—gravelly sand, fiber-reinforced gravelly sand, cement-stabilized gravelly sand, and cement-stabilized and fiber-reinforced gravelly sand. The impacts of compaction methods, standard and modified Proctor tests, and the curing time of the stabilized samples were also established.

## 2. Materials and Methods

### 2.1. Materials

Studies were conducted on two research samples of non-cohesive soil and cement-stabilized soils, as well as their mixtures with different quantities of polypropylene fibers.

Figure 1 illustrates the grain-size distribution curves of the different samples of the tested soils based on sieve analyses according to the EN 933-1 standard [31]. The estimated material is a coarse soil, with sand as its primary fraction and gravel as its secondary fraction. The tested soil is gravelly sand (grSa) in accordance with the EN ISO 14688-1 standard [32]. The values of the coefficient of uniformity (*C*_U_) and the coefficient of curvature (*C*_C_), calculated based on the grain-size distribution curves of the research samples I and II, were 5.45 and 0.87, and 5.04 and 0.66, respectively. The tested soil can be assessed as poorly graded based on the EN ISO 14688-2 standard [33]. The tested soil meets the standard requirements [34,35] for subbase or base materials with a lower percentage of fine fractions, which is suitable in frost areas.

Gravelly sand is a Pleistocene glaciofluvial soil characterized by a variability in the relief surface related to the high dynamics of the sedimentary environment and the variety of mineral compositions of post-glacial soils. Gravelly sand consists of well-rounded quartz crumbles, as well as angular grains, with a considerable amount of lytic particles and feldspars [28,36].

The compaction parameters, optimum water content (w_opt_), and maximum dry density (*ρ*_d max_) were established in accordance with the standard Proctor (SP) and modified Proctor (MP) methods, following the EN 13286-2 standard [37]. The compaction curves of the two different research samples of gravelly sand, along with saturation lines, are shown in Figure 2. In the cases of both research samples, for the soil compacted by means of a higher compaction energy, the degree of saturation, $(S_{r}$) is slightly higher than that for the soil compacted by means of a lower compaction energy. Sample I, with slightly better graining parameters, (*C*_U_)and (*C*_C_) is characterized by a greater density obtained at a lower water content in both compaction methods, standard and modified, than that of sample II.

The hydraulically bound mixture of soil and cement was created with the addition of 42.5R Portland cement in the amount of 1.5% of the dry mass of the cement to the dry mass of the soil in an examined sample. An attempt was also made to test the soil samples and the soil–cement mix with dispersed reinforcement in the form of polypropylene fibers. Thus, 18 mm-long fibers were used, which were added at amounts of 0.1%, 0.2%, and 0.3% in relation to the dry mass of the compacted soil. The polypropylene fibers are shown in Figure 3. The soil and polypropylene fiber mixtures were mixed by the means of a laboratory mechanical stirrer, which is of great importance for the homogeneity of the tested samples.

The compaction parameter (*ρ*_d max_ and *w*_opt_), void ratio at maximum compaction (*e*), and specific dry density (*ρ*_s_) values of all the tested materials are presented in Table 1.

For both compaction methods, it can be noted that the value of the void ratio, (*e*) decreases with an increase in the cement addition to the mixture. The specific dry density slightly increases with the addition of the cement to the mixture. However, the addition of the polypropylene fibers generally does not influence the specific dry density because of their low mass. The maximum dry density increases, while the void ratio decreases after the addition of the polypropylene fibers to the gravelly sand or to the sand and cement mixture. Generally, the optimum water content decreases thereafter.

### 2.2. Methods

#### 2.2.1. California Bearing Ratio Test

The California Bearing Ratio (*CBR*) is defined as follows:(3)$$CBR=\frac{p}{p_{s}}100\%$$
where *p* is the unit load used to press a standardized piston in a soil to a specific depth at a rate of 1.25 mm/min, and $p_{s}$ is the unit load needed to press the piston at the same rate into the same depth of a crushed rock at standard compaction.

The *CBR* laboratory tests were conducted on the samples of the gravelly sand and its mixtures with 1.5% cement and/or various percentages of polypropylene fibers in the amounts of 0.1%, 0.2%, and 0.3%. The percentage represents the dry mass of the additive per the dry mass of the soil in a specimen. For hydraulically bound material, dry soil was mixed with the fibers and cement by means of a laboratory mechanical stirrer; then, water was added to gain a moisture content relating to *w*_opt_ (see Table 1). The samples were compacted at optimum water contents using the standard Proctor (SP) and the modified Proctor (MP) methods in *CBR* molds. The *CBR* tests were performed on the samples directly after compaction (hydraulically unstabilized) and on the samples compacted and cured for 7 and 28 days at constant humidity and a temperature of 20 °C to avoid drying. The tests presented in this paper were performed only on unsoaked samples to enable comparisons of the test conditions during the *CBR* and resilient modulus tests.

The samples were loaded following the ASTM D1883 standard [38], with a recommended load of 2.44 kPa (4.54 kg) in the static penetration tests. Figure 4a shows a loaded sample in a *CBR* mold prior to testing. The larger *CBR* value was accepted as a result calculated based on the piston resistance at a given depth: 2.5 or 5.0 mm. The results obtained were collected using a computer program.

In accordance with the requirements of the EN 13286-47 standard [39], immediate bearing index tests were also carried out, i.e., *CBR* tests without a load on the sample. The tests of the hydraulically bound mixtures in this case were carried out no later than 90 min after mixing.

#### 2.2.2. Resilient Modulus in Cyclic Triaxial Apparatus

The resilient modulus, (*M_r_*) is expressed by the following formula:(4)$$M_{r}=\frac{\sigma_{cyclic}}{\varepsilon_{r}}$$
where $\sigma_{cyclic}$ is the amplitude of the applied cyclic axial stress, and $\varepsilon_{r}$ is the relative resilient (recovered) axial strain.

Laboratory tests were executed in the cyclic triaxial test apparatus on the gravelly sand and its mixtures with 1.5% cement and/or various percentages of polypropylene fibers in the amounts of 0.1%, 0.2%, and 0.3%. The percentage represents the dry mass of the additive per the dry mass of the soil in an examined sample. At first, the dry components were mixed using the laboratory mechanical stirrer; then, water was added to obtain a moisture content corresponding to *w*_opt_ (see Table 1). The cylindrical specimens, of 70 mm ID and 140 mm high, were compacted by impact in three layers in a bipartite mold to obtain the maximum dry density values found using the standard Proctor (SP) and the modified Proctor (MP) tests (Table 1), and they were then relocated to the triaxial chamber. The $M_{r}$ tests were conducted on the specimens directly after compaction (hydraulically unstabilized) and after 7 and 28 days of curing at a constant temperature and humidity. An image of a sample prepared for testing is shown in Figure 5.

In the cyclic triaxial apparatus, the confining pressure and axial load were put on pneumatically. The machine used repeated cycles of the haversine-shaped load pulse, where the load pulse lasted 0.1 s, and the rest period lasted 0.9 s. The variations in the sample height in the course of the tests were measured using external LVDT displacement transducers. The test settings and the results found were controlled and saved by a computer program. An image of a sample in the triaxial chamber prior to the cyclic test is shown in Figure 4b.

The specimens were exposed to cyclic loading in order to establish the resilient modulus $M_{r}$ according to the AASHTO T307 standard [7]. Table 2 describes the sequence 0–15 data for the subgrade material. Sequence “0” is the conditioning of the specimen. The number of cycles for this sequence was 500–1000 cycles; for all following sequences, it was constant and equal to 100. In the subsequent fifteen sequences, the confining pressure ranged from 20.7 to 137.9 kPa, and the maximum axial stress ranged from 20.7 to 275.8 kPa. The $M_{r}$ value was calculated for sequences from 1 to 15 as the average value of the previous five cycles of each load sequence.

## 3. Results and Discussion

Table 3 presents the California Bearing Ratio test results obtained for the gravelly sand and the sand with the 1.5% cement addition. Two different research samples taken from the same deposit were tested. All types of specimens were tested alone or improved by polypropylene fibers, with a length of 18 mm, in the amounts of 0.1%, 0.2%, and 0.3% in a mass ratio related to the dry soil mass. The specimens were compacted by means of the standard or modified Proctor method at the optimum water content. The hydraulically bound specimens were tested immediately after compaction or after 7 or 28 days of curing. The samples were tested unloaded (immediately after compaction) or loaded at 2.44 kPa.

The gravelly sand without any improvement showed relatively high *CBR* values, which were higher under the minimum load of 2.44 kPa than unloaded. These values are 25.9–37.0% and 56.4–90.2% for the standard and modified Proctor compaction methods, respectively. Higher *CBR* values were obtained for sample I, which was characterized by a slightly better particle size distribution (higher values of graining coefficients, *C*_U_ and *C*_C_) and, hence, a higher density after compaction. The addition of 0.1% polypropylene fibers caused an almost 2-fold increase in the *CBR* value of the samples compacted using the standard method. Increasing the amount of fibers to 0.2 and 0.3% reduced the *CBR* value to the value obtained without the addition of fibers. The addition of 0.1% fibers to the samples compacted using the modified method resulted in an approximate 10% decrease in the *CBR* value, and the increase in the amount of fibers to 0.2 and 0.3%—further resulted in a slight decrease in the *CBR* value. A reduction in the *CBR* value after the addition of the fibers occurred despite the improvement in the compaction of the reinforced sand (see Table 1).

The results of the non-cohesive soil reinforced with fibers differ from those of previous tests of the shear strength of reinforced granular soils [30], where good reinforcement results were achieved; however, Yetimoglu and Salbas [40] also found a decrease in the strength of sand with the addition of fibers, especially under higher stress. They found that the reinforcement changed the brittle medium into a more plastic one. An improvement in mechanical properties with an increase in fiber addition was observed in [41] in the case of *CBR* results; however, these results were found for reinforced clayey soil.

The addition of cement in the amount of 1.5% to the gravelly sand improved the *CBR* values of the cured samples, achieving the values of 68.1–143.9% and 150.9–171.3% for the standard compaction and 82.7–198.7% and 183.6–281.0% for the modified ones after 7 and 28 days of sample curing, respectively. The samples with the cement addition, tested directly after compaction, generally obtained lower *CBR* values than those without the cement addition. The 0.3% addition of fibers to the cement-stabilized samples improved their *CBR* values immediately after compaction and after curing for 7 and 28 days. The improvement depended on the duration of the sample curing and the compaction method—in the case of a sample compacted using the standard method and tested directly after compaction, the increase in *CBR* was about 100%. With the time of curing, this increase reduced to about 50%. In the case of samples compacted using the modified method, this increase reduced from about 50% to 4% after sample curing.

The increase in the *CBR* value after the addition of a hydraulic stabilizer is well-documented in the literature, mainly for lime-stabilized cohesive soils [42]. Cement-stabilized clayey soil with polypropylene fibers was tested by Tang et al. [43]. They stated that increasing the fiber content could weaken the brittle behavior of cemented soil by increasing the peak axial stress and by decreasing the stiffness and the loss of the post-peak strength. Wang et al. [44] also stated that the fibers in lime-stabilized clay mainly acted on the ductility lifting thereof. The values of UCS of fiber-reinforced lime, and cement-stabilized clayey soil increase with an increase in the curing time [43,45].

Table 4 presents the results of the cyclic triaxial tests of the resilient modulus. The gravelly sand and the sand improved with cement and/or polypropylene fibers were tested. A set of samples was prepared for the tests in accordance with description in Table 1.

The resilient modulus values found for the unbound gravelly sand in all tests were zero for both the natural and fiber-reinforced soils. The specimen was damaged by shearing after only the first test sequence of about 600 cycles (Figure 6a). The soil fiber reinforcement enabled only one extra sequence of tests to be carried out and the extension of the test to approximately 700 load cycles (Figure 6b). Several tests did not allow for the determination of the $M_{r}$ value of the gravelly sand without the addition of cement, even after fiber reinforcement.

The cement additive had a significant impact on the increase in the resilient modulus value. The 1.5% cement addition significantly increased the soil resistance to cyclic loads. The resilient modulus increased from 0 to an average of 340 MPa after 28 days of curing. In the case of the gravelly sand, the soil compaction method was less important, as the results achieved for both compaction methods were similar, although they were generally higher for the specimens compacted using the modified method. For the stabilized samples after 28 days of curing, $M_{r}$ values equal to 245–349 MPa and 313–513 MPa were obtained for the standard and modified compaction methods, respectively. The effect of the curing time was noticeable, and a longer curing duration resulted in a generally higher $M_{r}$. The exceptions were the samples compacted using the standard method with the addition of fibers, where extending the sample curing time from 7 to 28 days worsened the $M_{r}$ value. This phenomenon should be explored in the future by means of SEM image analyses. The samples stabilized with the 1.5% cement addition achieved resilient modulus values of 140–324 MPa after 7 days of curing and 245–513 MPa after 28 days of curing. Higher values were obtained for sample I, a soil with better graining and compaction. The addition of the fibers to the samples stabilized with 1.5% cement increased the $M_{r}$ value. Figure 6c presents a gravelly sand sample improved by the addition of fibers and cement and subjected to the resilient modulus test.

Figure 7 shows the different failure modes of the samples damaged during the resilient modulus test or after the cyclic test and quick shearing. The gravelly sand sample was sheared during the $M_{r}$ cyclic test, and it is characterized by the barreling mode with a single inclined failure plane (Figure 7a). After the $M_{r}$ test and quick shearing, the cement-bound soil failure mode was vertical through the failure plane (Figure 7b), as in the case of brittle material, whereas the plane of destruction in the soil sample improved by the cement and fiber addition was close to horizontal (Figure 7c). The horizontal failure plane indicates that the fiber-reinforced material entered its plastic zone. Both cement-stabilized samples were tested after seven days of curing.

The test results indicate a positive impact of the cement addition and the hardening time on the cyclic resistance of non-cohesive soil. In the case of cement-stabilized cohesive soils, an increase in the $M_{r}$ value has also been found after adding cement and extending the curing time [46,47], as in the case of lime-stabilized cohesive soils [48]. However, the resilient response of cohesive soils treated with lime is visible directly after sample preparation for uncured soils [49]. The fiber reinforcement of lime-treated cohesive soil has also been found to improve the $M_{r}$ value [50].

Figure 8 illustrates the statistical dependence of $M_{r}$ on *CBR* found for bound and unbound gravelly sand, as a dependence of the cyclic and static material stiffness, for the different prepared specimens. For all test results, the statistically valid relationship of $M_{r}=f\left(CBR\right)$ was found, where the coefficient of determination *R*^2^ = 0.6988, and the standard error of estimate SEE = 73.23 MPa. Equation $M_{r}=1.30\;CBR-31.78$ explains 69.9% of the variance in the value of the $M_{r}$ statistic for the whole data set. As the $M_{r}$ value of the unbound samples was zero regardless of the compaction method and the addition of dispersed reinforcement, which had a significant impact on the *CBR* value, it was decided to investigate the dependence of $M_{r}$ on *CBR* with the exclusion of these points. Once these points were excluded, a statistically invalid relationship of $M_{r}=0.31\;CBR+180.57$ was found, where the coefficient of determination *R*^2^ = 0.1145, and the standard error of estimate SEE = 62.06 MPa.

The relationship of $M_{r}=f\left(CBR\right)$ found for all data sets, although assessed as being statistically significant, is not useful from an engineering point of view due to its large standard error of estimate, because $M_{r}=1.30\;CBR-31.78\pm 73.23\;\mathrm{MPa}$. Overall, the unbound gravelly sand as a poorly graded material is non-resistant to cyclical interactions regardless of the compaction method. The fines content, (*f*_C_) interpreted according to ASTM D653 [51] as soil content with a grain size of less than 0.075 mm, was equal to zero for the tested sand. The lack of fines caused a complete lack of coherence between the grains and the damage by shearing at the beginning of the cyclic test. The resistance to cyclic interactions was not significantly improved, even after the addition of the polypropylene fibers as dispersive reinforcement. For gravelly sand with a lack of fines, the formula $M_{r}=f\left(CBR\right)$ should not be used.

The relationship of $M_{r}=f\left(CBR\right)$ found for the bound samples with the 1.5% cement addition is statistically invalid, i.e., cyclic and static material stiffnesses are independent, so $M_{r}=f\left(CBR\right)$ should also not be used.

It should be noted that Reference [52] found low $M_{r}$ values (but non-zero values) for compacted coarse granular material, but AASHTO T307 standard [7] procedure was not used. The test was limited to 300 cycles, which could certainly affect the test result. In the authors research, gravelly sand could not be damaged by up to about 600 load cycles (Figure 6a). The medium to well-graded crushed limestone aggregates were characterized by high values of $M_{r}$, greater for material with better graining [20]. The crushed gravel exhibited resistance to cyclic loading assessed using a summary of resilient modulus values taking into account modeled values based on the regulations of the National Cooperative Highway Research Program [53].

## 4. Conclusions

Based on the test results of compacted gravelly sand in a natural state and that improved by the addition of polypropylene fibers and/or cement, the following conclusions were drawn:The gravelly sand, characterized by a lack of fines and compacted at the optimum water content to the maximum dry density, was sheared during the resilient modulus test after about 600 cycles of loading regardless of whether the standard or modified compaction method was used. The fiber reinforcement of the gravelly sand slightly improved cyclic resistance, but the samples were sheared after about 700 loading cycles. A resilient modulus value could not be achieved for the unbound gravelly sand.The 1.5% cement addition increased the resilient modulus of the gravelly sand, which was 140–324 MPa after 7 days of curing and 245–513 MPa after 28 days of curing. Therefore, the curing time also impacted the resilient modulus value. The addition of fibers to the gravelly sand stabilized with 1.5% cement increased $M_{r}$ but changed the material behavior at failure from brittle to plastic.The gravelly sand revealed high *CBR* values: 25.9–37.0% and 56.4–90.2% for the standard and modified compaction methods, respectively. An addition of 0.1% fibers doubled the *CBR* values of the samples compacted using the standard method; however, increasing the addition to 0.2 and 0.3% reduced the *CBR* value to the value found for the samples without fibers. The addition of 0.1% fibers to the modified compacted samples resulted in an approx. 10% decrease in the *CBR* value, and the increase in the fibers to 0.2 and 0.3% resulted in further minor drops in the *CBR* value. The reduction in the sand stiffness may have been caused by its plasticization upon the addition of the fibers.The addition of 1.5% cement improved the *CBR* value of the gravelly sand, with the values reaching 68.1–143.9% and 150.9–171.3% for the standard compaction and 82.7–198.7% and 183.6–281.0% for the modified compaction after 7 and 28 days of curing, respectively. The addition of 0.3% fibers to the cement-stabilized soil improved its *CBR* value directly after compaction and curing, which depended on the duration of the curing time and the compaction method used.Even though the obtained relationship of $M_{r}=f\left(CBR\right)$, found for all data sets, was evaluated as being statistically significant, it is not efficient from an engineering point of view. The unbound gravelly sand was non-resistant to cyclic loading regardless of the compaction method used. The above dependency of the bound samples is statistically invalid, i.e., cyclic and static material stiffnesses are quite independent. In the authors’ opinion, the *CBR* and resilient modulus values should not be compared due to the various ways of loading samples during tests—static and cyclic loading dependent on the stress state, especially in the case of non-cohesive materials.

## Figures and Tables

**Figure 1 materials-16-00417-f001:**
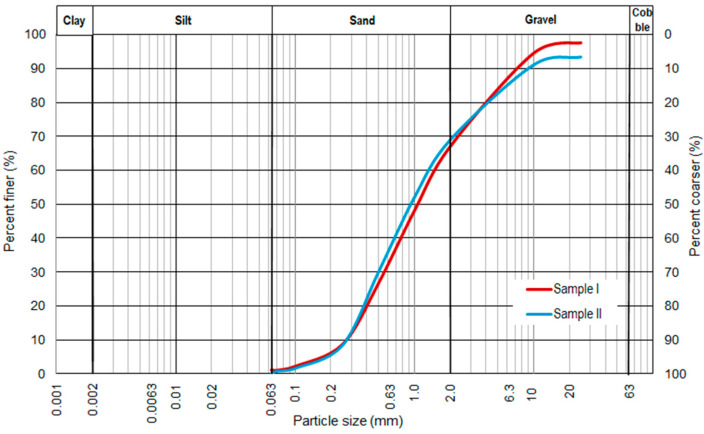
Grain-size distribution curves for research samples of non-cohesive soil.

**Figure 2 materials-16-00417-f002:**
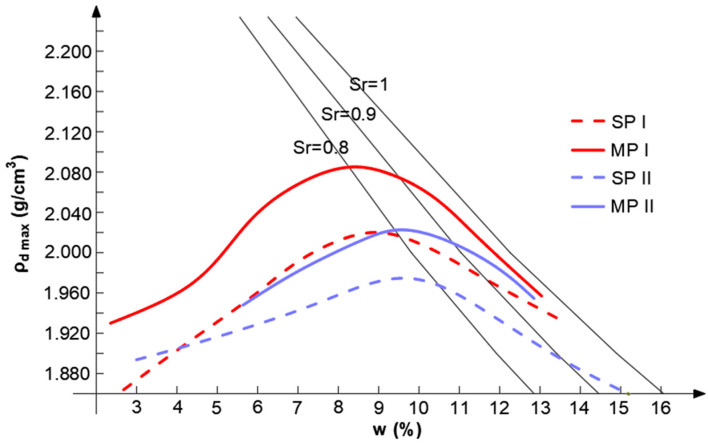
Compaction curves obtained using the standard Proctor (SP) and modified Proctor (MP) methods for research samples I and II.

**Figure 3 materials-16-00417-f003:**
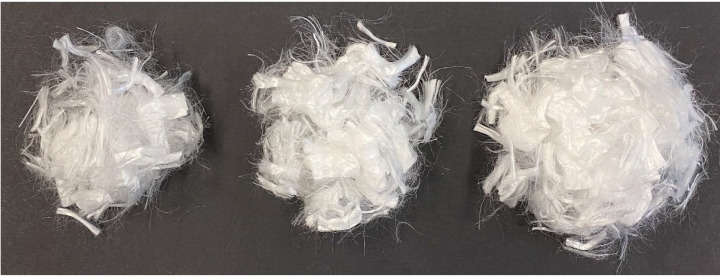
The 18 mm-long polypropylene fibers added at mass ratios of 0.1%, 0.2%, and 0.3% to the dry mass of the compacted soil.

**Figure 4 materials-16-00417-f004:**
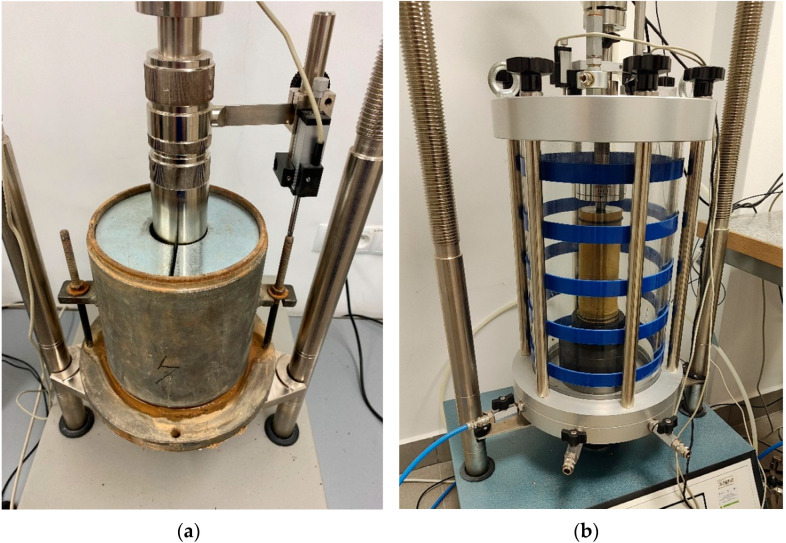
Image of a loaded sample prior to the test: (**a**) in the *CBR* mold, (**b**) in the chamber of the cyclic triaxial apparatus.

**Figure 5 materials-16-00417-f005:**
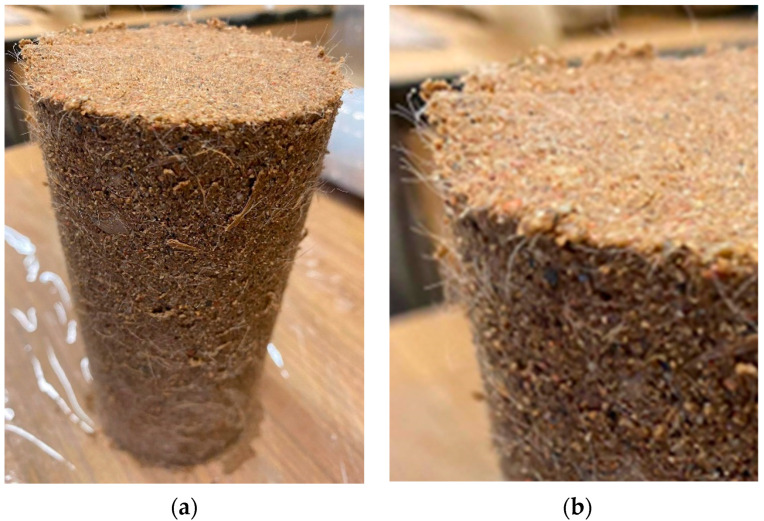
Image of a sample (grSa + 1.5C + 0.3%F 18 mm) compacted via SP method: (**a**) sample directly after compaction, (**b**) magnification of the sample with a visible homogeneous distribution of reinforcement.

**Figure 6 materials-16-00417-f006:**
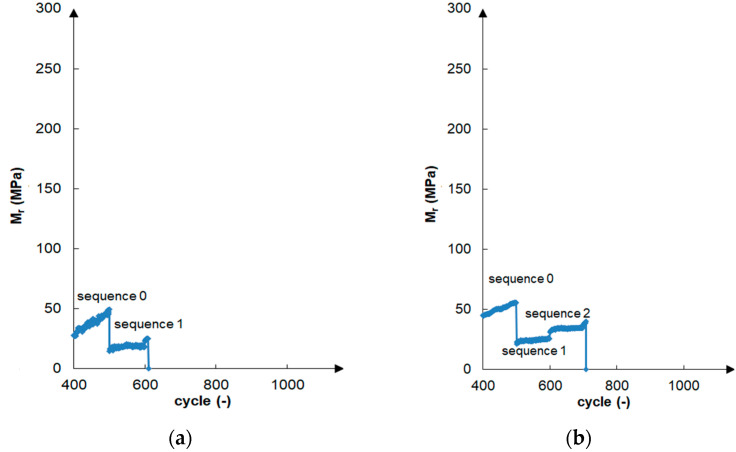
Variations in resilient modulus for the following samples: (**a**) grSa compacted via the SP method damaged in sequence “1”, (**b**) grSa + 0.2%F 18 mm compacted via the SP method damaged in sequence “2”, (**c**) grSa + 1.5%C + 0.2%F 18 mm compacted via the SP method and cured for 7 days tested in all test cycles (see Table 2).

**Figure 7 materials-16-00417-f007:**
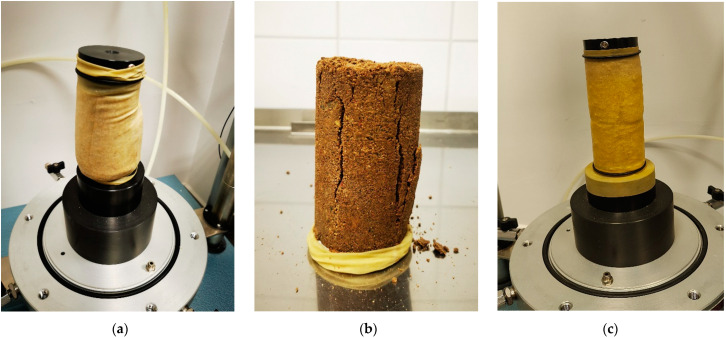
Typical failure modes of samples in the $M_{r}$ tests: (**a**) grSa compacted via SP method damaged during the $M_{r}$ test (sequence 1); (**b**) grSa + 1.5C compacted via MP method, cured for 7 days, after cyclic test and quick shearing; (**c**) grSa + 1.5C + 0.3%F 18 mm compacted via SP method, cured for 7 days, after cyclic test and quick shearing.

**Figure 8 materials-16-00417-f008:**
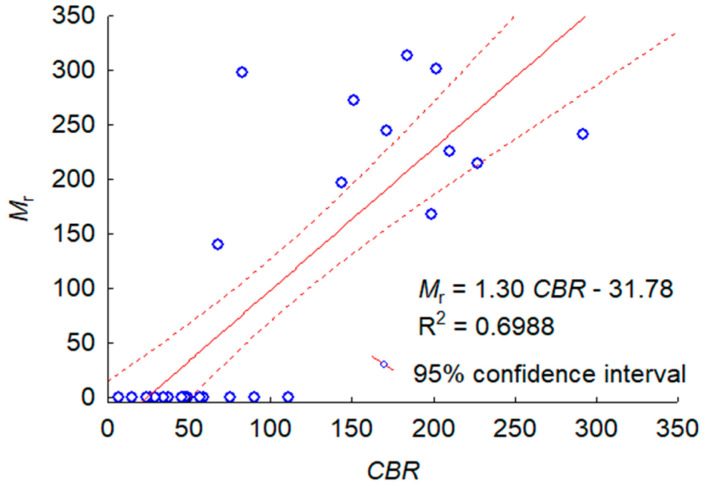
$M_{r}$ values vs. *CBR* values obtained for all tests along with 95% confidence intervals.

**Table 1 materials-16-00417-t001:** Geotechnical properties of tested materials.

Sample Number	Material	Compaction Method						*ρ*_s_ (g/cm^3^)
		Standard Proctor			Modified Proctor			
		*w*_opt_ (%)	*ρ*_d max_ (g/cm^3^)	*e* (–)	*w*_opt_ (%)	*ρ*_d max_ (g/cm^3^)	*e* (–)	
I	grSa	9.00	2.020	0.31	8.50	2.085	0.27	2.65
	grSa + 1.5%C	8.90	2.120	0.25	8.40	2.130	0.25	2.66
II	grSa	9.70	1.974	0.34	10.00	2.022	0.31	2.65
	grSa + 0.1%F 18 mm	9.80	2.003	0.33	8.80	2.104	0.27	2.65
	grSa + 0.2%F 18 mm	8.00	2.054	0.30	7.50	2.158	0.24	2.65
	grSa + 0.3%F 18 mm	7.70	2.060	0.30	7.00	2.160	0.24	2.65
	grSa + 1.5%C	9.50	2.010	0.32	9.50	2.070	0.29	2.66
	grSa + 1.5%C + 0.2%F 18 mm	7.90	2.066	0.29	7.00	2.183	0.23	2.66
	grSa + 1.5%C + 0.3%F 18 mm	8.00	2.060	0.30	7.80	2.124	0.26	2.66

C—cement addition, F—fiber addition.

**Table 2 materials-16-00417-t002:** Examination sequences for base or subbase material [7].

Sequence Number	Confining Pressure (kPa)	Max Applied Axial Stress (kPa)	Cyclic Stress $\sigma_{cyclic}$ (kPa)	Number of Load Applications
0	103.4	103.4	93.1	500–1000
1	20.7	20.7	18.6	100
2	20.7	41.4	37.3	100
3	20.7	62.1	55.9	100
4	34.5	34.5	31.0	100
5	34.5	68.9	62.0	100
6	34.5	103.4	93.1	100
7	68.9	68.9	62.0	100
8	68.9	137.9	124.1	100
9	68.9	206.8	186.1	100
10	103.4	68.9	62.0	100
11	103.4	103.4	93.1	100
12	103.4	206.8	186.1	100
13	137.9	103.4	93.1	100
14	137.9	137.9	124.1	100
15	137.9	275.8	248.2	100

**Table 3 materials-16-00417-t003:** California bearing ratio test results.

Sample Number	Material	Curing Time (Days)	*CBR* (%)			
			Standard Proctor		Modified Proctor	
			Unloaded	Loaded 2.44 kPa	Unloaded	Loaded 2.44 kPa
I	grSa	0	29.5	37.0	71.9	90.2
	grSa + 1.5%C	0	−	7.1	−	14.8
		7		68.1		82.7
		28		171.3		183.6
II	grSa	0	19.4	25.9	52.7	56.4
	grSa + 0.1%F 18 mm	0	−	46.0	−	49.4
	grSa + 0.2%F 18 mm	0	−	34.2	−	47.8
	grSa + 0.3%F 18 mm	0	−	28.7	−	45.8
	grSa + 1.5%C	0	27.9	24.1	55.5	75.6
		7		143.9		198.7
		28		150.9		281.0
	grSa + 1.5%C + 0.3%F 18 mm	0	−	59.7	−	111.3
		7		202.1		210.2
		28		227.3		290.7

**Table 4 materials-16-00417-t004:** Resilient modulus test results.

Sample Number	Material	Curing Time (Days)	*M_r_* (MPa)	
			Standard Proctor	Modified Proctor
I	grSa	0	0	0
	grSa + 1.5%C	0	0	0
		7	324	217
		28	349	513
	grSa + 1.5%C	0	0	0
		7	140	298
		28	245	313
II	grSa	0	0	0
	grSa + 0.1%F 18 mm	0	0	0
	grSa + 0.2%F 18 mm	0	0	0
	grSa + 0.3%F 18 mm	0	0	0
	grSa + 1.5%C	0	0	0
		7	197	168
		28	272	353
	grSa + 1.5%C + 0.2%F 18 mm	0	0	0
		7	271	215
		28	202	230
	grSa + 1.5%C + 0.3%F 18 mm	0	0	0
		7	301	226
		28	214	241

## Data Availability

The data presented in this study are available on request from the corresponding authors.

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
