# Peer review of "Estimation of Stiffness of Non-Cohesive Soil in Natural State and Improved by Fiber and/or Cement Addition under Different Load Conditions"

_materials, 2023, doi:10.3390/ma16010417_

Round 1
Reviewer 1 Report
See comments in the attached file

Author Response
Dear Sir,
I would like to thank for all the remarks on our manuscript. The paper after reorganizing and revision is more comprehensible.
I am sending along my paper reorganized according to the Reviewers’ suggestions ̶ new text is in red.
Yours faithfully,
Katarzyna Zabielska-Adamska
Reviewer 2 Report
The paper presents the impact of fibres and cement additives on both the CBR and the resilient modulus. The introduction to the problem, rationale and literature review are well written, informative and concise.
The methodology is clearly described and informative and the testing well conducted. The results appear appropriate and again are well explained. The attempt to relate CBR to Mr for the final results may possibly have had more success if the stabilisation methods were looked at separately but this may have been done and not reported on, but this is a very minor point when the standard error means that the relationship won’t be of engineering use.
Nice to see the limitations of the work being discussed. Overall the authors should be congratulation on a well-presented, interesting paper. Some very minor comments are in the pdf attached.

Author Response
Dear Sir,
I would like to thank for all remarks on our manuscript. The paper after reorganizing and revision is more comprehensible.
I am sending along my paper reorganized according to the Reviewers’ suggestions ̶ new text is in red.
In reply to the individual comments:
- For preparing samples of cyclic triaxial test, the authors did not mention how can they control all samples with a similar condition, for example the same void ratio, or relative density?
The sentence describing the sample preparation was changed on:
Lines 211-213: “The cylindrical specimens, of 70 mm ID and 140 mm high, were compacted by impact in three layers in a bipartite mold to obtain the values of maximum dry densities found by the standard Proctor (SP) and the modified Proctor (MP) tests (Table 1), and next relocated to triaxial chamber.”
- Why did the authors use 1.5% Portland cement and range of amount of fiber 0.1-0.3%?
The using 1.5% of cement has already been explained in the text as:
Lines 109-114: “The soil was tested as unbound and hydraulically bound with a 1.5% cement addition, selected as minimum amounts that could improve the resilient properties of the tested soil. As the previous experience of the authors [28,29] has shown that the cyclic triaxial test can destroy unbound non-cohesive soil, it was also decided to test soil samples reinforced with 18 mm long polypropylene fibers to improve tested material [30].”
The amount of polypropylene fibers was selected based on the literature. In the case of the sands, 0.1-0.5% polypropylene fibers were added most often in relation to the dry mass of the soil. Since it was wanted to show the effect of fiber addition, it was decided that the additive would be from 0.1-0.3%. It has been shown that additives greater than or equal to 0.2% worsen the mechanical properties of the tested soil.
- The author should discuss the data in Table 3-4:
- Table 3: when increasing % of fiber to 0.1% the CBR of SP samples still increased, but % of fiber going up to 0.2-0.3%, the CBR value decreased?
The problem has been already discussed in the text as:
Lines 250-257: “The addition of 0.1% polypropylene fibers caused an almost 2-fold increase in the CBR value of samples compacted with the standard method. Increasing the amount of fibers to 0.2 and 0.3% reduced the CBR value to the value obtained without the addition of fibers. The addition of 0.1% of the fibers to the samples compacted by the modified method resulted in an approximate 10% decrease in the CBR value, and the increase in the amount of fibers to 0.2 and 0.3% − further slight decreases in the CBR value. The reduction in the CBR value after the addition of the fibers occurred despite the improvement in compaction of the reinforced sand (see Table 1).”
The phenomenon was explained based on literature concerned shear strength tests when it was stated that “the reinforcement changed the brittle medium into a more plastic one” – Lines 258-264.
- Table 4: Why when adding fiber, the Mr of 1.5%C after 7 days increasing, but the Mr of 1.5%C after 28 days decreasing compared to non-adding fiber samples?
It was added to the text:
Lines 300-302: “The exceptions were samples compacted with the standard method with the addition of fibers, where extending the sample curing time from 7 to 28 days worsened the Mr. This phenomenon should be explained in the future by means of SEM image analysis.
Yours faithfully,
Katarzyna Zabielska-Adamska
Reviewer 3 Report
The paper "Estimation of stiffness of non-cohesive soil in natural state and improved by fiber and/or cement addition in different load conditions" presents a relevant theme and within the scope of this journal, and can be considered after some corrections suggested below:
(a) The abstract is generally well written, however in terms of content it is generic, i.e., the authors lack an in-depth study of the quantitative results of this research;
(b) Scientific innovation is limited in the introduction of the paper, the authors must go deeper and detail what this research differs from countless others that exist on this topic, this must be evidenced together with the objectives at the end of the introduction;
(c) The state of the art of the evaluated topic needs to be improved by the authors, note that some topics are absent and need to be known with current research, such as: 10.1590/1807-1929/agriambi.v24n3p187-193; 10.1016/j.cscm.2022.e00943; 10.1016/j.cscm.2021.e00837.
(d) At the end of the introduction, the paper does not clearly show its objectives (general and specific), in addition to the innovation and motivation of this research, this should be evidenced;
(e) “The cement additive has a significant impact on the increase in the value of the resilient modulus. In the case of the gravelly sand, the method of soil compaction was less important, and the results obtained for both compaction methods are similar, although generally higher for specimens compacted by the modified method.” These excerpts should be better explained by the authors.
(f) “Lack of fines causes a complete lack of coherence between grains and damage by shearing at the beginning of the cyclic test. The resistance to cyclic interactions was not improved significantly even after addition of polypropylene fibers as dispersive reinforcement. For gravelly sand with a lack of fines formulas ?? = ?(???) should not 351 be used.” These excerpts should be better explained by the authors.
(g) The conclusion must be reformulated, it is generally very extensive.
Author Response
Dear Sir,
I would like to thank for all the remarks on our manuscript. The paper after reorganizing and revision is more comprehensible.
I am sending along my paper reorganized according to the Reviewers’ suggestions ̶ new text is in red.
In reply to the individual comments:
(a) The abstract is generally well written, however in terms of content it is generic, i.e., the authors lack an in-depth study of the quantitative results of this research.
Unfortunately, the authors cannot deepen the analysis of the text in the abstract because the abstract has the maximum possible number of words − 200.
(b) Scientific innovation is limited in the introduction of the paper, the authors must go deeper and detail what this research differs from countless others that exist on this topic, this must be evidenced together with the objectives at the end of the introduction;
and
(d) At the end of the introduction, the paper does not clearly show its objectives (general and specific), in addition to the innovation and motivation of this research, this should be evidenced.
Now lines 106-120 are: “The authors concluded from their previous research that it may not be possible to provide the relationship for non-cohesive soil because the cyclic triaxial test can destroy unbound non-cohesive soil [28,29]. Thus, the aim of the study is to compare the stiffness tested under various load conditions on a typical non-cohesive soil used for the construction of a road base and subbase – postglacial gravelly sand. The static and cyclic loads, represented by CBR and resilient modulus tests, were considered. The soil was tested as unbound and hydraulically bound with a 1.5% cement addition, selected as minimum amounts that could improve the resilient properties of the tested soil. As the previous experience of the authors has shown that non-cohesive soil can be sheared during the cyclic triaxial test, it was also decided to test soil samples reinforced with 18 mm long polypropylene fibers to improve tested material [30]. Thus, the behavior of four varied materials will be considered – gravelly sand, fiber reinforced gravelly sand, cement-stabilized gravelly sand, and cement-stabilized and fiber reinforced gravelly sand. The influence of compaction by the standard and modified Proctor methods and the curing time of stabilized samples were also established.”
(c) The state of the art of the evaluated topic needs to be improved by the authors, note that some topics are absent and need to be known with current research, such as: 10.1590/1807-1929/agriambi.v24n3p187-193; 10.1016/j.cscm.2022.e00943; 10.1016/j.cscm.2021.e00837.
According to the authors, none of the indicated literature items is related to the subject of the article, even indirectly.
The first paper (10.1590/1807-1929/agriambi.v24n3p187-193):
Afonso R. Azevedo, Markssuel T. Marvila, Euzébio B. Zanelato, Jonas Alexandre, Gustavo C. Xavier & Daiane Cecchin (2020). Development of mortar for laying and coating with pineapple fibers. Revista Brasileira de Engenharia Agrícola e Ambiental Campina Grande, Vol.24, 187-193.
The paper concerns addition of NaOH treated pineapple fibers to cement mortars. None of cement mortar mechanical properties were tested.
The second paper (10.1016/j.cscm.2022.e00943):
Mustafa Fahmi Hasan, Hanifi Canakci (2022). Physical-mechanical assessment of full-scale soil-cement column constructed in clayey soil. Case Studies in Construction Materials, Vol. 16, e00943.
The paper concerns jet grouting columns constructed in clayey soil. Tested properties of columns depends on pressure and rotation. Authors also predict the diameter of column.
The third paper (10.1016/j.cscm.2021.e00837):
Prinya Chindaprasirt, Arkhom Sriyoratch, Anukun Arngbunta, Panatchai Chetchotisak, Peerapong Jitsangiam, Apichit Kampala (2022). Estimation of modulus of elasticity of compacted loess soil and lateritic-loess soil from laboratory plate bearing test. Case Studies in Construction Materials, Vol. 16, e00837.
The paper concerns other parameters (static), other laboratory device (plate bearing test) and other soil (loess, lateritic-loess). The soil was not improved by cement or polypropylene fibers. In this case, at least the article concerns soil research in the laboratory, which is not a sufficient condition to quote it.
The authors cited as many as 53 items, and the article is not a review article. In the opinion of the authors, the quoted number of publications is still very large.
(e) “The cement additive has a significant impact on the increase in the value of the resilient modulus. In the case of the gravelly sand, the method of soil compaction was less important, and the results obtained for both compaction methods are similar, although generally higher for specimens compacted by the modified method.” These excerpts should be better explained by the authors.
Now lines 295-302 are: “The cement additive has a significant impact on the increase in the value of the resilient modulus. Already 1.5% cement addition significantly increased the soil resistance to cyclic loads. The resilient modulus rises from zero to an average of 340 MPa after 28 days of curing. In the case of the gravelly sand, the method of soil compaction was less important, and the results obtained for both compaction methods are similar, although generally higher for specimens compacted by the modified method. For the stabilized samples after 28 days of curing, the Mr was obtained equal to 245-349 MPa and 313-513 MPa, at the standard and modified compaction methods, respectively.”
(f) “Lack of fines causes a complete lack of coherence between grains and damage by shearing at the beginning of the cyclic test. The resistance to cyclic interactions was not improved significantly even after addition of polypropylene fibers as dispersive reinforcement. For gravelly sand with a lack of fines formulas ?? = ?(???) should not be used.” These excerpts should be better explained by the authors.
In the authors opinion these sentences should be read together with above text. In this case everything is understandable. Please read lines 349-359: “The relationship found for all data set, although assessed as statistically significant, is not useful from an engineering point of view due to such a large standard error of estimate, because of . Above all, unbound gravelly sand as a poorly graded material is non-resistant to cyclical interactions, independently on the compaction method. The content of fines, fC, interpreted in accordance with ASTM D653 [51] as the soil content with a grain size less than 0.075 mm, is equal to zero for tested sand. Lack of fines causes a complete lack of coherence between grains and damage by shearing at the beginning of the cyclic test. The resistance to cyclic interactions was not improved significantly even after addition of polypropylene fibers as dispersive reinforcement. For gravelly sand with a lack of fines formulas should not be used.”
(g) The conclusion must be reformulated, it is generally very extensive.
The authors do not understand this remark. According to the authors, the conclusions should refer to all the achievements of the work - it was successful. In the authors opinion conclusion should have general statement based on the paper results – it has been done. As some reviewers / researchers believe that conclusions should refer to numerical values, this has also been done.
The conclusions are extensive because many studies have been carried out on four different soils compacted by two methods.
Yours faithfully,
Katarzyna Zabielska-Adamska

Round 2
Reviewer 3 Report
The paper is suitable for publication.